# Guided self-help treatment for children and young people with threshold and subthreshold eating disorders: A pilot study protocol

**Emily Davey**[1]*, **Rachel Bryant-Waugh**[2,3,4], **Sophie Bennett**[1,5], **Nadia Micali**[1,6], **Julian Baudinet**[2,3,4], **Sam Clark-Stone**[7], **Roz Shafran**[1]

1 UCL Great Ormond Street Institute of Child Health, University College London, London, United Kingdom, 2 Maudsley Centre for Child and Adolescent Eating Disorders, South London and Maudsley NHS Foundation Trust, London, United Kingdom, 3 Department of Child and Adolescent Psychiatry, Institute of Psychiatry, Psychology & Neuroscience, King's College London, London, United Kingdom, 4 Centre for Research in Eating and Weight Disorders (CREW), Institute of Psychiatry, Psychology & Neuroscience, King's College London, London, United Kingdom, 5 Department of Psychology, Institute of Psychiatry, Psychology & Neuroscience, King's College London, London, United Kingdom, 6 Mental Health Services of the Capital Region of Denmark, Center for Eating and Feeding Disorders Research (CEDaR), Ballerup Psychiatric Centre, Copenhagen, Denmark, 7 The Eating Disorders Service, Gloucestershire Health and Care NHS Foundation Trust, Cheltenham, United Kingdom

* emily.davey.21@ucl.ac.uk

## Abstract

### Background

Prompt access to evidence-based treatment for children and young people with eating disorders is important for outcomes, yet the gap in service provision remains pervasive. Record levels of young people are waiting for eating disorder treatment and access to care is limited. Guided self-help interventions that are brief and require minimal clinician support have the potential to meet the unprecedented demand for treatment quickly and effectively.

### Objective

To examine the feasibility, acceptability and proof of concept of a novel, CBT guided self-help intervention for children and young people with threshold and subthreshold eating disorders.

### Methods

A single-arm, proof-of-concept pilot study of the CBT guided self-help intervention will be conducted. Children and young people (aged 11–19) with threshold and subthreshold eating disorders will receive a self-help intervention covering the core components of CBT, supported by 8 weekly guidance sessions delivered remotely. Clinical outcomes (eating-related psychopathology and associated impairment, changes in weight, depression, anxiety, and behavioural difficulties) will be assessed at baseline and post-intervention (12 weeks). Feasibility and acceptability of the intervention will be measured using various outcomes,

**Data Availability Statement:** No datasets were generated or analysed during the current study. All

relevant data from this study will be made available upon study completion.

**Funding:** This research is funded by a Child Health Research PhD Studentship at UCL Great Ormond Street Institute of Child Health. The funders did not and will not have a role in the study design, data collection and analysis, decision to publish, or preparation of the manuscript.

**Competing interests:** The authors have declared that no competing interests exist.

including adherence to, and engagement with the intervention, rates of recruitment and retention, measure completion and treatment satisfaction. Qualitative data will also be collected for future intervention refinement.

## Discussion

If the intervention is shown to produce clinical benefits in this pilot study, a fully powered randomised pilot study will be warranted with the ultimate goal of increasing access to psychological treatment for children and young people threshold and subthreshold eating disorders.

## Administrative information

This study protocol (S1 File) adheres to the guidelines outlined in the Standard Protocol Items: Recommendations for Interventional Trials (SPIRIT) checklist for trial protocols [1, 2] which can be found in S1 Checklist. The numbers in parentheses in this protocol correspond to the item numbers in the SPIRIT checklist. The order of items has been modified to group similar items.

## Introduction

### Background and rationale (6)

Eating disorders have serious adverse physical and psychological consequences [3]. Individuals with eating disorders have greater mortality rates and poorer quality of life compared to those with other psychiatric conditions [4, 5]. The economic impact of eating disorders is significant, primarily due to elevated rates of hospitalisation, outpatient care and visits to emergency departments [6]. Eating disorders typically begin during adolescence [7], and are prevalent among those who present to child and adolescent mental health services in the UK [8]. Given the illness burden and associated costs [9], there is a critical need for accessible, effective treatments for children and young people with clinical eating disorders and those whose eating difficulties do not meet the diagnostic threshold for an eating disorder (i.e., they are subthreshold) but whose food, weight and shape-related concerns and behaviours negatively impact their physical, mental or emotional health.

Evidence-based, specialist psychological interventions have good empirical support for treating eating disorders in this population [10]. However, access to evidence-based care has long been challenging [11], and this has been exacerbated further by the COVID-19 pandemic [12]. The COVID-19 pandemic has had profound effects on young people with eating disorders, with worsened symptom severity and an increased incidence of new-onset diagnoses [13]. Child and adolescent eating disorder services in the UK have seen nearly a twofold increase in the number of referrals [8], and it is now estimated that 59.4% of young people aged 17–19 have disordered eating and 12.5% have a diagnosable eating disorder [14]. National Health Services (NHS) have struggled to meet this increased demand for treatment, with only 79% of urgent and 82% of routine cases seen within the one week and four week respective standards (a target of 95%) during October—December 2022 [15]. This is concerning given that delays in intervention can lead to a protracted illness course and poorer health outcomes [16, 17]. Although numerous geographical, financial and patient-associated barriers may contribute to this treatment gap [18], a fundamental issue is that the demand for eating

disorder treatment surpasses the availability of resources [19, 20]. Less resource-intensive interventions that are clinically effective and broadly accessible must be leveraged to meet this unprecedented demand for treatment quickly and effectively [21].

Guided self-help interventions are an obvious choice to meet such a demand. They are the first step treatment for adults with bulimia nervosa and binge eating disorder [22], and have proven efficacy for children and young people with anxiety and depression [23, 24], which often co-occur with eating disorders [25]. However, guided self-help interventions are not used routinely in child and adolescent eating disorder services in the UK due to a lack of sufficient research on their efficacy [26]. The existing but limited research indicates the potential effectiveness of guided self-help interventions for this patient group. Lock et al. [27] found an online guided self-help version of family-based treatment (GSH-FBT) for parents of adolescents with anorexia nervosa led to clinical improvements, including weight gain and reduced eating disorder psychopathology. In their randomised controlled trial, Schmidt et al. [28] found that CBT guided self-care resulted in a more rapid reduction of binge eating and proved to be more cost-effective than family therapy for adolescents with bulimia nervosa and related disorders. Despite these encouraging findings, both interventions are still within the realm of research and are not readily accessible for children and young people with threshold and subthreshold eating disorders in the UK.

A recent consensus statement by UK experts highlights the suitability of guided self-help interventions for various eating disorder presentations across all age groups, contingent upon close monitoring of medical and psychiatric risks [21]. The statement emphasises the importance of improving the provision, accessibility, and adherence to these interventions. Adherence to treatment recommendations and subsequent good clinical outcomes are more likely if individuals perceive the intervention as acceptable [29, 30]. Moreover, ensuring the acceptability of remote delivery for self-help interventions is crucial, especially considering young people's consistent preference for face-to-face specialist eating disorder treatment over that of their parents/carers [31–33].

## Objectives (7)

This study aims to examine the feasibility, acceptability and proof-of-concept of a CBT, guided self-help intervention for children and young people (aged 11–19 years) with threshold and subthreshold eating disorders.

## Study design (8)

This proof-of-concept pilot study will use a single-arm, pre-post design with an embedded qualitative and quantitative process evaluation. Children and young people (aged 11–19) with threshold and subthreshold eating disorders will receive a CBT self-help intervention and 8 x 30-minute weekly guidance sessions delivered remotely. Participants, and their parents/carers, will complete assessments at baseline and post-intervention (12 weeks after baseline).

# Methods: Participants, interventions and outcomes

## Study setting (9)

Recruitment will take place at participating specialist eating disorder services in England. To see the participating sites, please visit https://doi.org/10.1186/ISRCTN16038125.

## Eligibility criteria (10)

Individuals will be eligible to participate in this study if they meet the below eligibility criteria:
 **Inclusion criteria for children and young people.**

a. Aged 11–19 years

b. Assessment conducted by the referring Eating Disorders Service and/or screening assessment (ChEDE-Q8) indicates young person meets diagnostic criteria for anorexia nervosa, bulimia nervosa, binge eating disorder, other specified feeding or eating disorder (OSFED; atypical anorexia nervosa, bulimia nervosa of low frequency and/or limited duration or binge eating disorder of low frequency and/or limited duration), or has subthreshold but impairing symptoms related to these diagnoses

c. Is a UK resident

d. Has a parent/carer who is willing to take part in the study

**Exclusion criteria for children and young people.**

e. Does not speak or understand English sufficiently well to access the intervention

f. Has an intellectual disability at a level that impedes their ability to access the intervention

g. Assessment conducted by the referring Eating Disorders Service and/or screening assessment indicates young person has an eating disorder that does not meet the inclusion criterion (b) above such as avoidant restrictive food intake disorder (ARFID) and other feeding and eating disorders

h. Assessment conducted by the referring Eating Disorders Service and/or screening assessment indicates acute risk deemed unsuitable for the trial due to the clinical need for immediate and/or specialist intervention (e.g., rapid weight loss, very low mood, high medical or psychiatric risk, acute suicidality, recurrent or potentially life limiting self-harm and/or significant safeguarding concerns)

i. If prescribed psychotropic medication, the dosage must have remained stable for the preceding two months

j. Currently receiving other overlapping psychological support/intervention

k. Does not have access to a laptop or smartphone which they can use for guidance sessions

## Informed consent and assent (26a)

Written informed consent will be taken via an online consent form. The order of referral procedures and the consent process is as follows:

1. Potential participants will be referred to the study by a member of their clinical care team at the specialist eating disorder service.

2. The study team will contact the participant (or parent/carer when appropriate) to provide more information and details of the consent process.

   Parents/carers and children and young people (aged $\geq$ 16) will be asked to provide informed consent to participate in the study. Children and young people (aged < 16) will be asked to provide informed assent, in addition to the consent of their parent/carer.

## Confidentiality (27)

Identifiable information pertaining to the participant is essential for participant registration. Confidentiality will be upheld, except in cases where a participant discloses information that suggests a risk of harm to themselves or others. In the event of a risk and/or safeguarding issue

being identified, the research team will inform the participant's GP and other relevant professionals involved in their care (e.g., the referring Eating Disorders Service).

### Intervention (11a)

The intervention is described in accordance with the Template for Intervention Description and Replication (TIDieR) checklist ([34]; see S2 Checklist). The study acronym is SPICE, which stands for Short Psychological Intervention for Children and adolescents with Eating disorders. The intervention is a modular, guided self-help intervention, based on principles of CBT. The intervention was developed using a common elements approach across the three pillars of evidence-based practice: the best available scientific research, clinical expertise and patient preferences [35]. This involved a systematic review to identify evidence-based, CBT-self-help interventions for eating disorders [26], from which the treatment components were extracted. A qualitative study was also conducted to explore the preferences of key stakeholders (i.e., young people with lived experience, parents/carers, and clinicians) regarding the content, structure, and delivery mode of self-help interventions for children and young people with eating disorders. A common elements approach was then undertaken to establish the shared elements across the evidence-based CBT interventions and the key areas considered important for incorporation by stakeholders. Intervention content was derived from publicly available resources, with permission granted by the original authors for adaptation and credit given for their contributions. Intervention materials have been reviewed by Patient and Public Involvement (PPI) representatives and revisions were made based on their feedback.

The current treatment manual is comprised of eight treatment modules (Table 1). The intervention covers the core components of CBT for eating disorders, including psychoeducation, reducing eating disorder behaviours, improving body image, addressing shape checking and avoidance, challenging negative thoughts, regulating emotions and preventing relapse [36, 37]. It also covers improving self-esteem and navigating social media. The intervention is delivered through an interactive Portable Document Format (PDF) workbook. Each module has its own PDF workbook and accompanying home practice tasks.

Participants will be offered eight weekly guidance sessions over the telephone/videocall (mode dependent on participant preference) with a guide, each lasting up to 30 minutes. The young person's parent/carer is encouraged to attend these guidance sessions. Participants will be asked to read the relevant module and complete between-session tasks prior to each guidance session. Participants will keep the intervention materials after the study has ended, providing a resource that they can return to at a later date. The self-help intervention will be supported by the first author (ED), a postdoctoral researcher in child and adolescent eating disorders. ED will receive one day of online training and weekly supervision from RS. During the guidance sessions, the guide will assume a facilitative role. The aim of their guidance is to enhance motivation, troubleshoot problems that arise and refer participants to the intervention content to enhance knowledge and skills usage.

### Criteria for discontinuing or modifying allocated interventions (11b)

As weekly symptom tracking measures are being used, the guide and their supervisors will be able to quickly detect significant deterioration. If no progress and/or deterioration is seen over the first six sessions [38, 39], and this is not attributable to other known factors (e.g., a life event), the study team will discuss possible discontinuation with the family and referral to other more appropriate sources of support.

**Table 1. Treatment modules which comprise CBT guided self-help for children and young people with eating disorders.**

| | Module | Description |
|---|---|---|
| 1. | Understanding my eating difficulties | The aim of this module is to provide key information about eating disorders, including what keeps them going, and to help the young person think about what is keeping their eating difficulties going. The young person will also identify goals for the intervention. |
| 2. | Eating more regularly | The aim of this module is to help the young person understand the relationship between what they are eating and their energy levels. It will also introduce ways to improve the structure of their eating and the types of food that they eat. |
| 3. | Reducing dieting | The aim of this module is to help the young person identify and challenge any strict diet rules that are keeping their eating difficulties going, including rules around when to eat, what to eat and how much to eat. |
| 4. | Doing things differently | The aim of this module is to provide the young person with some strategies to reduce and manage weight control behaviours, such as self-induced vomiting, laxatives and exercising excessively. |
| 5. | Body image and social media | The aim of this module is to provide some strategies to help the young person tackle concerns around their body image. It will also discuss the role of social media in body image and ways to use social media in a more positive way. |
| 6. | Learning to feel good about myself | The aim of this module is to provide the young person with some effective ways to improve their self-esteem. |
| 7. | Managing emotional triggers | The aim of this module is to explain the link between events, emotions and eating. It will help them to consider healthier ways to cope, including how to solve day-to-day problems. |
| 8. | Planning for the future | The aim of this module is to help the young person maintain the progress they have made. It will support the young person to develop a plan for managing slips or setbacks which may happen in the future. |

The order in which modules are completed is decided in collaboration with the young person based upon their personalised formulation and treatment goals.

Participation in this study is entirely voluntary. Participants have the right to discontinue the trial intervention at any time, without incurring a penalty or loss of entitlement to benefits.

## Strategies to improve adherence to intervention protocols (11c)

ED will follow a protocol to guide the delivery of the intervention. They will also receive weekly supervision from RS throughout the research and intervention process. Guidance sessions will be audio/video recorded and played in supervision to ensure adherence to the agreed protocol. Procedural fidelity will also be monitored through the completion of adherence to the protocol checklists after each guidance session.

## Relevant concomitant care permitted or prohibited during the trial (11d)

Participants who are currently undergoing psychological treatment for an emotional or behavioural problem will not be permitted to participate in the trial. If the individual has been prescribed psychotropic medication, the dosage must have remained stable for the past two months to participate in the trial.

## Provisions for post-trial care (30)

There are no plans for ancillary or post-trial care. Any significant risk issues that arise will be shared with the participants' GP and any other clinician involved in their care (e.g., the referring Eating Disorders Service). Families will be signposted to appropriate services and sources of support as necessary.

## Outcome measures (12)

The EDE-Q global score is the primary outcome; all other outcomes described below are secondary.

**Eating disorder psychopathology.**  *Eating disorder examination questionnaire (EDE-Q;* [40]*) and parent eating disorder examination questionnaire (PEDE-Q;* [41]*)*. The EDE-Q is a 28-item self-report measure of eating disorder attitudes and behaviours over the past month. Eating disorder symptoms will be measured using the EDE-Q global score, the mean of the subscale scores (dietary restraint, eating concern, weight concern and shape concern). Scores range from 0 to 6, with higher scores indicating more severe symptomatology. The EDE-Q has shown good reliability and convergent validity [42].

The PEDE-Q is a 29-item parent-report measure that assesses parental perceptions of the severity of their child's eating disorder symptoms during the preceding 28 days. Its content and scoring system mirror those of the EDE-Q. The PEDE-Q demonstrates good psychometric properties and offers incremental information that can contribute to a more comprehensive assessment of eating disorders in adolescents [43].

*The clinical impairment assessment (CIA;* [44]*)*. The CIA is a 16-item self-report questionnaire that measures psychosocial impairment associated with eating disorder symptoms over the past 28 days. A CIA global score is calculated as a severity index (ranging from 0 to 48), where higher scores denote greater functional impairment. The CIA has well-documented validity and reliability [45]. The CIA has been modified to make it applicable to children and young people (e.g., item 4 now asks about school performance instead of work performance).

## Percentage median body mass index (%mBMI)

Participants' body weight and height will be measured using the relevant items on the EDE-Q [40]) so that %mBMI ((actual BMI ÷ 50th centile BMI) x 100) can be determined.

**Depression and anxiety symptomatology.**  *The revised child anxiety and depression scale (RCADS;* [46]*)*. The self-report and parent-report versions of the RCADS will be used to assess changes in the young person's symptoms of DSM-defined anxiety disorders and major depression. The RCADS is a 47-item questionnaire measuring five anxiety subtypes and depression symptoms with robust psychometric properties [47, 48]. *t*-scores will be calculated based on the child's school year and gender, based on developer norms [47].

*The strengths and difficulties questionnaire (SDQ;* [49]*)*. The self-report and parent-report versions of the SDQ will be used to assess the young person's psychological wellbeing. The SDQ is a 25-item questionnaire comprising five subscales, each containing five items. These subscales assess conduct problems, emotional problems, hyperactivity, peer problems, and prosocial behavior. The total difficulties score is obtained by summing the scores from the conduct, emotional, hyperactivity, and peer problems subscales, resulting in a possible range of 0 to 40. Higher scores indicate greater difficulties. An impact supplement scale consisting of 5 items assesses the functional impairment of the identified problems. The reliability and validity of the SDQ is well established [50, 51].

**Feasibility and acceptability.**  Feasibility and acceptability of the intervention and study procedures will be examined through rates of referral and uptake, session attendance, intervention completion/attrition and measure completion. In addition, participant satisfaction with the intervention will be assessed using a seven-item questionnaire adapted from Creswell et al. [52]. Participants, and their parents/carers, will be asked to rate seven areas of satisfaction on five-point Likert scales: change in the young person's disordered eating, satisfaction with the programme, the overall help received, number of guidance sessions, length of guidance sessions, overall satisfaction and whether they would recommend the approach to others.

| | Enrolment | Baseline | Intervention | Post-intervention |
|---|---|---|---|---|
| **Enrolment:** | | | | |
| Informed assent (CYP <16 years)<br>Informed consent (CYP 16+ years)<br>Informed consent (parent/carers) | X | | | |
| Demographic information | X | | | |
| CHEDE-Q8 | X | | | |
| **Intervention:** | | | | |
| CBT guided self-help for CYP with eating disorders | | | X | |
| **Assessments:** | | | | |
| **Young person** | | | | |
| DAWBA | | X | | |
| EDE-Q | | X | | X |
| CIA | | X | | X |
| RCADS | | X | | X |
| SDQ | | X | | X |
| Acceptability questionnaire | | | | X |
| Optional qualitative interview | | | | X |
| **Parent/carers** | | | | |
| PEDE-Q | | X | | X |
| RCADS | | X | | X |
| SDQ | | X | | X |
| Acceptability questionnaire | | | | X |
| Optional qualitative interview | | | | X |
| **Session-by-session measures:** | | | | |
| ED-15-Y / ED-15-P | | | X | |
| GBOs (CYP and parent reported) | | | X | |
| SDQ SxS (CYP and parent reported) | | | X | |

**Fig 1. The time schedule of enrolment, interventions, and assessments for participants.** CYP = children and young people.

Families will also be invited to participate in an optional qualitative interview to explore their experience of receiving the intervention.

## Participant timeline (13)

A visual summary of the timeline and assessment points for participants can be found in Fig 1.

## Sample size (14)

As formal sample size calculations are unnecessary for proof-of-concept studies [53], the number of children and young people recruited to receive the CBT guided self-help intervention is not pre-specified. However, the ORBIT model states that proof-of-concept studies can have small sample sizes because clinical, not statistical, benefit is sought [53].

## Recruitment (15)

Participants will be referred to the study by the collaborating Eating Disorders Services. Participants can provide their consent to be contacted and patient details will be given to the research team. Alternatively, participants can scan a QR code on the participant information sheet and share their contact details with the research team via an online form. The research team will contact the participant to talk through the study procedures and to answer any questions. If interested, participants can then consent to take part and complete a screening questionnaire. If the screening questionnaire indicates that the participant meets the eligibility criteria, they will be invited to enrol in the study.

# Methods: Data collection, management and analysis

After providing informed consent, participants will be allocated a unique, sequential study identifier which will be used to label all study data. Data collected at baseline and post-intervention will be collected using REDCap electronic data capture tools hosted on the UCL Data Safe Haven. REDCap is a secure web platform for building and managing online databases and surveys [54]. The UCL Data Safe Haven is a technical environment for receiving, handling and storing sensitive data securely. It has been certified to the ISO27001 information security standard and conforms to NHS Digital's Information Governance Toolkit.

## Plans for assessment and collection of outcomes (18a)

Assessment will be at screening, baseline and post-intervention (12 weeks after baseline completion).

**Screening and eligibility assessment.**   All families who consent/assent to participate will complete a short screening questionnaire to determine their eligibility for participation. This includes demographic information about the young person and their parent/carer, history of young person's psychological support (including prescribed psychotropic medication) and a question around the young person's current self-harm and suicidality. It also involves the young person completing the Child Eating Disorder Examination Questionnaire– 8-items (ChEDE-Q8; [55]) to assess their eating disorder symptomatology.

**Baseline assessment.**   Young people will complete a battery of baseline measures, including the EDE-Q, CIA, RCADS and SDQ. Parents will complete parent-report versions of the PEDE-Q, RCADS and SDQ. Participants will also be asked to complete the Development and Wellbeing Assessment (DAWBA [56]) to enable an understanding of comorbid diagnoses. The DAWBA will be piloted with two families in the first instance. If the DAWBA is considered too burdensome, it will no longer be administered to families. If the family have already completed the DAWBA during an assessment with their eating disorder service, this data will be accessed to prevent duplication of measures.

**Post-intervention assessment.**   Post-intervention assessment will take place 12 weeks after completion of the baseline measures. This will include the EDE-Q [PEDE-Q], CIA, RCADS and SDQ, as well as an acceptability questionnaire.

**Qualitative interview.**    All young people and parents who consent to participate in the study will be invited to take part in an optional qualitative interview about their experiences of the study and the intervention. Interviews will cover what families found helpful/unhelpful about the intervention, views on the mode, content and structure of the treatment and suggestions for improvement. Families who dropout of the study will be invited to take part in an interview, unless they withdraw consent to be recontacted.

**Session-by-session measures.**    Families will also complete weekly questionnaires for intervention progress monitoring, including the Eating Disorder-15 for Youth / Parents (ED-15-Y/P; [57, 58], Goal Based Outcomes (GBOs; [59]) and SDQ Session by Session (SDQ SxS; [60]).

## Plans to promote participant retention and complete follow-up (18b)

Participants will be encouraged to prioritise attendance at guidance sessions and the completion of post-intervention measures. Participants retain the right to withdraw from the study at any time, and such a decision will have no effect on their clinical care. Participants who discontinue treatment should remain in the study for the purpose of follow-up and data analysis. Such participants will be asked, without obligation, to complete post-intervention measures. They will also be asked to indicate their ongoing consent/assent to being contacted for the qualitative interview. Data will be analysed on an intention to treat and last measure carried forward basis, such that withdrawal from the study does not result in the exclusion of the participant's data from analysis.

## Data management (19)

Upon study entry, each participant will be assigned a unique, sequential study identifier. Participants will be identified by this number on all study-related documentation throughout the course of the intervention and data analysis process. All study data will be appropriately secured to meet governance requirements for collecting personal data. Identifiable patient data will be managed in accordance with the Caldicott Principles and Data Protection Act 2018, which mandates data to be anonymised as soon as it is practical to do so.

## Statistical methods for analysing primary and secondary outcomes (20a)

Statistical analyses will be carried out using IBM SPSS Statistics 27. Descriptive statistics will be used to characterise the sample, examine participant flow through the study, assess attrition rates at each stage, and to evaluate satisfaction with the intervention. All open-text data will be collated. Due to the lack of control group, expected small sample size and consequently insufficient power to test for significance, only descriptive analyses will be performed on the primary and secondary outcomes.

**Baseline and post-intervention measures.**    Mean values and mean differences (MD) will be used to describe proof-of-concept for eating disorder psychopathology, %mBMI, depression, and anxiety from baseline and post-intervention. Data will be presented as either the percentage of change (for continuous variables) or absolute scores (for discrete variables).

**Session-by-session measures.**    Weekly questionnaires (ED-15-Y/P, GBOs and SDQ SxS) will be visually inspected and descriptive statistics will be provided.

**Qualitative interview.**    Qualitative data from the interviews with young people and their parents will be audio recorded, transcribed and analysed using reflexive thematic analysis [61]. Braun and Clarke's [62] guidance on improving the conduct and reporting of thematic analysis will be followed to ensure high quality analyses.

### Patient and public involvement (PPI)

Families with lived experience of eating disorders have been consulted at various stages throughout this project. PPI input has been invaluable, providing feedback on the participant information sheets, assessment protocol, intervention materials and plans for dissemination.

## Ethics and dissemination

### Research ethics approval (24)

This study has received ethical approval from the West of Scotland Research Ethics Committee 5 (approval number: 23/WS/0097), and approvals from the research and development departments at Great Ormond Street Hospital for Children NHS Foundation Trust, South London and Maudsley NHS Foundation Trust and Gloucestershire Health and Care NHS Foundation Trust.

### Plans for dissemination (31a)

Best-practice recommendations will be used to guide the dissemination process. The plan is to engage with children and young people, parents/carers, clinicians and policymakers to share study findings. The results will be published in peer-reviewed journals and presented at national and international conferences to reach academic and clinical audiences.

### Authorship (31b)

The International Committee of Medical Journal Editors' (ICMJE) recommendations for determining authorship will be followed.

## Discussion

To our knowledge, this will be the first study to evaluate a CBT guided self-help intervention in a transdiagnostic sample of children and young people with threshold and subthreshold eating disorders–a group which urgently need timely access to effective treatment [15]. The intervention protocol has been adapted from empirically supported cognitive behavioural treatment and developed in collaboration with families with lived experience.

This intervention is designed to be applicable to both those seeking treatment within NHS services as part of stepped care provision as well as those who do not seek services but prefer to use self-help materials, potentially with family rather than professional support. This is a small, proof-of-concept pilot study with no control or comparison groups. This will not allow comparative insights into whether any improvements are due to the CBT guided self-help intervention or the passing of time. As such, caution will be taken when interpreting the results. If the intervention achieves clinical benefits in this small, select sample, more rigorous pilot testing with a randomised design and larger, representative sample will be warranted [53].

## Supporting information

**S1 Appendix. Model consent form.**
(DOCX)

**S1 Checklist. SPIRIT checklist.**
(DOCX)

**S2 Checklist. TIDieR checklist.**
(DOCX)

**S1 File. Study protocol.**
(DOCX)

## Acknowledgments

The authors would like to acknowledge the contribution of families who participated in patient and public involvement activities, as well as colleagues across different NHS sites who have worked on this project. All research at Great Ormond Street Hospital for Children NHS Foundation Trust and UCL Great Ormond Street Institute of Child Health is made possible by the NIHR Great Ormond Street Hospital Biomedical Research Centre.

## Author Contributions

**Conceptualization:** Emily Davey, Rachel Bryant-Waugh, Sophie Bennett, Nadia Micali, Julian Baudinet, Sam Clark-Stone, Roz Shafran.

**Funding acquisition:** Emily Davey, Roz Shafran.

**Supervision:** Rachel Bryant-Waugh, Sophie Bennett, Nadia Micali, Roz Shafran.

**Writing – original draft:** Emily Davey, Roz Shafran.

**Writing – review & editing:** Emily Davey, Rachel Bryant-Waugh, Sophie Bennett, Nadia Micali, Julian Baudinet, Sam Clark-Stone, Roz Shafran.

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
