## [Decision Letter · Decision Letter 0]

2 Feb 2024

PONE-D-23-42547Guided self-help treatment for children and young people with threshold and subthreshold eating disorders: A pilot study protocolPLOS ONE

Dear Dr. Davey,

Thank you for submitting your manuscript to PLOS ONE. After careful consideration, we feel that it has merit but does not fully meet PLOS ONE’s publication criteria as it currently stands. Therefore, we invite you to submit a revised version of the manuscript that addresses the points raised during the review process.

Please address all the concerns made by reviewers. Despite they all agree with a Minor Revision, please consider thoroughly their suggestions and comments. As this revision process involved a statistical advisor, please refer especially to their comment for that methodological aspect. This does not exclude nor compete with the comments made by the other two reviewers, which should be considered and integrated in your final revised version of the manuscript.

We look forward to receiving your revised manuscript.

Kind regards,

Simone Varrasi

Academic Editor

PLOS ONE

4. We note that the original protocol that you have uploaded as a Supporting Information file contains an institutional logo. As this logo is likely copyrighted, we ask that you please remove it from this file and upload an updated version upon resubmission.

Reviewers' comments:

Reviewer's Responses to Questions

**Comments to the Author**

1. Does the manuscript provide a valid rationale for the proposed study, with clearly identified and justified research questions?

Reviewer #1: Yes

Reviewer #2: Yes

Reviewer #3: Yes

2. Is the protocol technically sound and planned in a manner that will lead to a meaningful outcome and allow testing the stated hypotheses?

Reviewer #1: Partly

Reviewer #2: No

Reviewer #3: Partly

3. Is the methodology feasible and described in sufficient detail to allow the work to be replicable?

Reviewer #1: Yes

Reviewer #2: Yes

Reviewer #3: Yes

4. Have the authors described where all data underlying the findings will be made available when the study is complete?

Reviewer #1: Yes

Reviewer #2: Yes

Reviewer #3: Yes

5. Is the manuscript presented in an intelligible fashion and written in standard English?

Reviewer #1: Yes

Reviewer #2: Yes

Reviewer #3: Yes

6. Review Comments to the Author

You may also provide optional suggestions and comments to authors that they might find helpful in planning their study.

Reviewer #1: This is a comprehensive and detailed protocol for a pilot study and process evaluation to assess the initial effectiveness, feasibility and acceptability of a newly developed CBT-based guided self-help intervention for a small group of children and young people (CYP) with a range of eating disorder diagnoses and severity. Recent consensus in programme-led interventions for young people with eating disorders, highlights the need for work in this area.

The manuscript meets the PLOS publication criteria including originality, clearly outlined methodology, presentation, writing style and clearly reported ethical approval, trial registry and reporting checklist.

Rationale and research questions clearly stated?

Excellent and succinct rationale for conducting the pilot study with clearly outlined aims. This could be strengthened further by linking to the recently published consensus statement for programme-led interventions to highlight the need, not only to improve provision of such interventions but also to optimise access and adherence

I the protocol technically sound to enable to test stated hypotheses?

The protocol is technically sound and presented in line with the SPIRIT checklist for clinical trials protocols.

However, I have slight concerns about the sample size of 10 CYP with a range of eating disorder presentations and the known drop-out rate in existing guided self-help interventions (circa 40%). This limits the conclusions the authors are able to make. Findings will be simply indicative as the authors acknowledge.

Is the methodology described in sufficient detail for replication?

The authors present the methods with an excellent level of detail and they are presented in line with SPIRIT checklist for clinical trial protocols, including important information such as strategies to improve adherence/ fidelity, criteria for discontinuation, managing risk.

Eligibility criteria: please state which screening tool you will use to determine whether an individual meets diagnostic criteria for an eating disorder

It would be useful to present details of the intervention using the TIDIER checklist

Outcome: Make clear which your primary outcome will be - EDEQ? And also whether you are interested in global score, cut-off or a particular behavioural feature? It would be useful to state this for each outcome measure in the text.

Have authors stated where data will be made available when complete?

This is a Protocol Paper therefore no data will be generated. The authors provide plans for dissemination in peer reviewed journals and academic conferences on completion of the study.

Is the manuscript written in an intelligible fashion?

The authors write in a clear and concise manner throughout

Additional comments:

In the Methods section of the abstract, please make clear the mode of guidance, as this is important (remotely)

In the abstract consider describing the study design as ‘open pilot trial and process evaluation’ so congruent with Study Design section in main text.

Study Design: Again, outline mode of guidance (remotely)

26a – simplify heading to ‘Informed consent and assent’ rather than a question

11a Intervention – it would be useful to provide details of the process of intervention development. Was it through principles of co-production? If so, how did you go about it?

Reviewer #2: You are planning a small sample size that will be recruiting from a variety of eating disorder types. Will your sample size be able to offer a sufficient evidence for your intervention's efficacy across these different diagnoses?

NICE guidelines recommend a range of treatment options depending on the classification of eating disorder. Do you expect your intervention to be suitable for all eating disorder types?

Reviewer #3: Important note: This review pertains only to ‘statistical aspects’ of the study and so ‘clinical aspects’ [like medical importance, relevance of the study, ‘clinical significance and implication(s)’ of the whole study, etc.] are to be evaluated [should be assessed] separately/independently. Further please note that any ‘statistical review’ is generally done under the assumption that (such) study specific methodological [as well as execution] issues are perfectly taken care of by the investigator(s). This review is not an exception to that and so does not cover clinical aspects {however, seldom comments are made only if those issues are intimately / scientifically related & intermingle with ‘statistical aspects’ of the study}. Agreed that ‘statistical methods’ are used as just tools here, however, they are vital part of methodology [and so should be given due importance]. I look at the manuscript in/with statistical view point, other reviewer(s) look(s) at it with different angle so that in totality the review is very comprehensive. However, there should be efforts from authors side to improve (may be by taking clues from reviewer’s comments). Therefore, please do not limit the revision only (with respect) to comments made here.

COMMENTS: There are very few issues/observations about which I have different opinion [mainly they are suggestions for further larger study (If the intervention is shown to be effective in this pilot study, a fully powered randomised controlled trial will be warranted with the ultimate goal of increasing access to psychological treatment for children and young people threshold and subthreshold eating disorders)]. Such observations/concerns are given below:

This study being ‘pilot’ (feasibility study) in nature, sample size is not a big issue. However, as mentioned in line 264 [This pilot study aims to recruit up to 10 eligible children and young people] the sample size intended to be used for this study is questionably small {as unlikely to get fair idea of attrition rates at each stage as said in line 338 and also unlikely to yield/produce an estimate of effect size which can be used to design a fully powered randomised controlled trial” as said in lines 386-88}. Further, though many things are ignored (loosely looked at / evaluated)] in case of ‘pilot studies’, methodological issues need to be very rigorously followed {in contrast with / despite often quote: “Pilot (Proof of Concept) studies typically involve a small number of subjects, as well as more latitude [i.e., leeway, freedom, liberty] in statistical requirements”}.

I request authors to read following note pasted from one famous standard textbook on ‘Medical Research Methodology’ [though I am sure that these learned authors already know these things] as it is very essential to keep the limitations in mind while interpreting results {note that I am not asking you to change the design of this pilot study}.

For a pilot study it is alright to have ‘single-arm design’, or it is alright when that is the only possibility’, however, it is very essential to keep the limitations in mind while interpreting results. Further, note that a classical/ideal clinical trial/study needs/requires a concurrently {but similarly} handled/treated appropriately selected/chosen control/comparison parallel group/arm.

Mind you further that any “Inferential statistics (i.e., hypothesis testing + estimation of CI) is built on the population model [which means the underlying assumption is that there is/are population(s) and we are dealing with random sample(s) drawn from that/those population(s)]. Although in clinical trial (involving at least two groups) we do not really deal with random samples (generally a non-probabilistic convenience sampling), ‘allocation’ to treatment groups is ‘randomly’ done which enable us to evoke the population model and we can use inferential statistics safely. But when there is only one group (so that there is no question of random allocation), with ‘non-random’ selection, it may be questionable to use inferential statistics even if you have two measurement sets as ‘pre-post’ or many repeated measurements or use ‘internal grouping for comparison.”

While planning larger study you may consider using ‘wait listed’ control group. Though the measures/tools used (in this pilot phase and few may be used in larger study) are appropriate, most of them are likely to yield data that are in ‘ordinal’ level of measurement [and not in ratio level of measurement for sure {as the score two times higher does not indicate presence of that parameter/phenomenon as double (for example, a Visual Analogue Scales VAS score or say ‘depression’ score)}]. Then application of suitable non-parametric (or distribution free) test(s) is/are indicated/advisable [even if distribution may be ‘Gaussian’ (also called ‘normal’)]. Agreed that there is/are no non-parametric test(s)/technique(s) available to be used as alternative in all situation(s), but should be used whenever/wherever they are available.

Therefore, in short use suitable non-parametric test(s)/technique(s) while dealing with data that are in ‘ordinal’ level of measurement even if [despite that] the distribution may be ‘Gaussian’. Testing ‘normality’ in sample [by using any normality test(s)} is not required/desired while dealing with data that are in ‘ordinal’ level of measurement [as most of the normality tests are not valid for ‘ordinal’ data].

On the backdrop of what is stated in ‘Abstract- Methods’ that “A mixed-method open pilot trial design will be used” please note that any regression techniques are not basically/originally developed for any sort of [between or within group(s)] comparison(s). Moreover, a brief note on intervention designed may be expected/desirable/appreciated, by most of the readers, I guess. Except these few points, this manuscript is ‘excellent’ and therefore only ‘Minor revision’ is recommended.

7. PLOS authors have the option to publish the peer review history of their article (what does this mean?). If published, this will include your full peer review and any attached files.

Reviewer #1: **Yes: **Dr Gemma D Traviss-Turner

Reviewer #2: **Yes: **Eleanor May Bowes

Reviewer #3: No

---

## [Author Response · Author response to Decision Letter 0]

15 Mar 2024

Reviewer 1

Comment 1: Excellent and succinct rationale for conducting the pilot study with clearly outlined aims. This could be strengthened further by linking to the recently published consensus statement for programme-led interventions to highlight the need, not only to improve provision of such interventions but also to optimise access and adherence.

Response 1: Thank you for your kind comments and for your suggestion to refer to the consensus statement. We have now added the following sentences to the introduction (page 7, lines 119-122): ‘A recent consensus statement by UK experts highlights the suitability of guided self-help interventions for various eating disorder presentations across all age groups, contingent upon close monitoring of medical and psychiatric risks (21). The statement emphasises the importance of improving the provision, accessibility, and adherence to these interventions.’

Comment 2: The protocol is technically sound and presented in line with the SPIRIT checklist for clinical trials protocols.

However, I have slight concerns about the sample size of 10 CYP with a range of eating disorder presentations and the known drop-out rate in existing guided self-help interventions (circa 40%). This limits the conclusions the authors are able to make. Findings will be simply indicative as the authors acknowledge.

Response 2: We agree that the conclusions that can be drawn from this study are somewhat limited due to the target sample size. However, we believe that publishing this study protocol even with a small sample size has value for open science. 

We have reframed this study as a proof-of-concept pilot study that seeks to determine whether more a rigorous randomised pilot trial is warranted. According to authors of the ORBIT model (Czajkowski et al., 2015), ‘sample size calculations are unnecessary’ for proof-of-concept studies, so we have decided not to pre-determine sample size for this study, as stated on page 16, lines 354-357. The model also states that sample sizes in proof-of-concept studies can be small because clinical, not statistical, benefit is sought (page 16, lines 356-357). The limitations will of course be considered in future work.

Comment 3: Eligibility criteria: please state which screening tool you will use to determine whether an individual meets diagnostic criteria for an eating disorder.

Response 3: To increase the reach of the intervention, individuals do not have to meet diagnostic criteria to be eligible for the study as stated on page 8, line 163: ‘has subthreshold but impairing symptoms related to these diagnoses’. Eligibility will be determined by the assessment conducted by the referring Eating Disorders Service and/or the screening tool, Child Eating Disorder Examination Questionnaire – 8 items (ChEDE-Q8). We have now clarified this by adding the following text to criterion B: Assessment conducted by the referring Eating Disorders Service and/or screening assessment (ChEDE-Q8) indicates young person meets diagnostic criteria for anorexia nervosa, bulimia nervosa, binge eating disorder, other specified feeding or eating disorder (OSFED; atypical anorexia nervosa, bulimia nervosa of low frequency and/or limited duration or binge eating disorder of low frequency and/or limited duration), or has subthreshold but impairing symptoms related to these diagnoses’ (page 8, lines 158-163).

Comment 4: It would be useful to present details of the intervention using the TIDIER checklist.

Response 4: Thank you for bringing the TIDIeR checklist to our attention. We have made changes to our description of the intervention in line with this checklist where applicable on pages 10-13). We have also attached the checklist as supplementary material which specifies where each item is located in the paper.

Comment 5: Outcome: Make clear which your primary outcome will be - EDEQ? And also whether you are interested in global score, cut-off or a particular behavioural feature? It would be useful to state this for each outcome measure in the text.

Response 5: We have now stated that the primary outcome will be the EDE-Q global scores (page 14, lines 292-293): ‘The EDE-Q global score is the primary outcome; all other outcomes described below are secondary.’ We have also provided further detail of the secondary outcome measures on page 15 (lines 319-330).

Comment 6: In the Methods section of the abstract, please make clear the mode of guidance, as this is important (remotely).

Response 6: We have now made it clear that the guidance sessions will be delivered remotely in the Methods section of the abstract: ‘supported by 8 weekly guidance sessions delivered remotely’ (page 2, line 16).

Comment 7: In the abstract consider describing the study design as ‘open pilot trial and process evaluation’ so congruent with Study Design section in main text.

Response 7: We have now described the study design as ‘a single-arm, proof-of-concept pilot study’ to make it congruent with the Study Design section in the main text (page 2, line 13).

Comment 8: Study Design: Again, outline mode of guidance (remotely).

Response 8: The study design now states the mode of guidance: ‘8 x 30-minute weekly guidance sessions delivered remotely’ (page 8, line 147).

Comment 9: 26a – simplify heading to ‘Informed consent and assent’ rather than a question.

Response 9: We have now simplified heading 26a to say ‘Informed consent and assent’ (page 9, line 193).

Comment 10: 11a Intervention – it would be useful to provide details of the process of intervention development. Was it through principles of co-production? If so, how did you go about it?

Response 10: Thank you for this suggestion. We have now provided further details of the systematic process used for intervention development on page 10 (lines 221-238): ‘This involved a systematic review to identify evidence-based, CBT-self-help interventions for eating disorders (26), from which the treatment components were extracted. A qualitative study was also conducted to explore the preferences of key stakeholders (i.e., young people with lived experience, parents/carers, and clinicians) regarding the content, structure, and delivery mode of self-help interventions for children and young people with eating disorders. A common elements approach was then undertaken to establish the shared elements across the evidence-based CBT interventions and the key areas considered important for incorporation by stakeholders. Intervention content was derived from publicly available resources, with permission granted by the original authors for adaptation and credit given for their contributions. Intervention materials have been reviewed by Patient and Public Involvement (PPI) representatives and revisions were made based on their feedback.’

Reviewer 2

Comment 1: You are planning a small sample size that will be recruiting from a variety of eating disorder types. Will your sample size be able to offer a sufficient evidence for your intervention's efficacy across these different diagnoses?

Response 1: Thank you for highlighting this. The intention of this pilot study is not to determine the efficacy of the current intervention. Rather, its purpose is to test whether the feasibility, acceptability and whether it produces clinical benefits that warrant further pilot testing. We have now reframed this study as a proof-of-concept pilot study which allow for small sample sizes because the focus is on clinical, not statistical, benefit. We have now made this clear on pages 8 (line 144), 16 (lines 354-357) and 23 (lines 503-508).

Comment 2: NICE guidelines recommend a range of treatment options depending on the classification of eating disorder. Do you expect your intervention to be suitable for all eating disorder types?

Response 2: Thank you for this question. Our formative qualitative work with young people, parents and healthcare professionals indicated that a transdiagnostic guided self-help intervention would be suitable for children and young people with eating disorders (paper submitted for publication). Consequently, the current intervention derives from CBT-E, a transdiagnostic treatment for all forms of eating disorder. 

In a recent consensus statement paper (Davey et al., 2023), there was consensus that suitability of guided self-help should not be pre-determined on diagnostic category alone. Hence, we have set out to recruit a varied sample of CYP with a range of eating disorders in this pilot study – the qualitative interviews will help us to ascertain its suitability across eating disorder types. 

We have now made reference to the consensus statement paper on page 7, lines 119-121: ‘A recent consensus statement by UK experts highlights the suitability of guided self-help interventions for various eating disorder presentations across all age groups, contingent upon close monitoring of medical and psychiatric risks.’

Reviewer 3 (statistical advisor)

Comment: This study being ‘pilot’ (feasibility study) in nature, sample size is not a big issue. However, as mentioned in line 264 [This pilot study aims to recruit up to 10 eligible children and young people] the sample size intended to be used for this study is questionably small {as unlikely to get fair idea of attrition rates at each stage as said in line 338 and also unlikely to yield/produce an estimate of effect size which can be used to design a fully powered randomised controlled trial” as said in lines 386-88}. Further, though many things are ignored (loosely looked at / evaluated)] in case of ‘pilot studies’, methodological issues need to be very rigorously followed {in contrast with / despite often quote: “Pilot (Proof of Concept) studies typically involve a small number of subjects, as well as more latitude [i.e., leeway, freedom, liberty] in statistical requirements”}.

I request authors to read following note pasted from one famous standard textbook on ‘Medical Research Methodology’ [though I am sure that these learned authors already know these things] as it is very essential to keep the limitations in mind while interpreting results {note that I am not asking you to change the design of this pilot study}.

For a pilot study it is alright to have ‘single-arm design’, or it is alright when that is the only possibility’, however, it is very essential to keep the limitations in mind while interpreting results. Further, note that a classical/ideal clinical trial/study needs/requires a concurrently {but similarly} handled/treated appropriately selected/chosen control/comparison parallel group/arm.

Mind you further that any “Inferential statistics (i.e., hypothesis testing + estimation of CI) is built on the population model [which means the underlying assumption is that there is/are population(s) and we are dealing with random sample(s) drawn from that/those population(s)]. Although in clinical trial (involving at least two groups) we do not really deal with random samples (generally a non-probabilistic convenience sampling), ‘allocation’ to treatment groups is ‘randomly’ done which enable us to evoke the population model and we can use inferential statistics safely. But when there is only one group (so that there is no question of random allocation), with ‘non-random’ selection, it may be questionable to use inferential statistics even if you have two measurement sets as ‘pre-post’ or many repeated measurements or use ‘internal grouping for comparison.”

While planning larger study you may consider using ‘wait listed’ control group. Though the measures/tools used (in this pilot phase and few may be used in larger study) are appropriate, most of them are likely to yield data that are in ‘ordinal’ level of measurement [and not in ratio level of measurement for sure {as the score two times higher does not indicate presence of that parameter/phenomenon as double (for example, a Visual Analogue Scales VAS score or say ‘depression’ score)}]. Then application of suitable non-parametric (or distribution free) test(s) is/are indicated/advisable [even if distribution may be ‘Gaussian’ (also called ‘normal’)]. Agreed that there is/are no non-parametric test(s)/technique(s) available to be used as alternative in all situation(s), but should be used whenever/wherever they are available.

Therefore, in short use suitable non-parametric test(s)/technique(s) while dealing with data that are in ‘ordinal’ level of measurement even if [despite that] the distribution may be ‘Gaussian’. Testing ‘normality’ in sample [by using any normality test(s)} is not required/desired while dealing with data that are in ‘ordinal’ level of measurement [as most of the normality tests are not valid for ‘ordinal’ data].

On the backdrop of what is stated in ‘Abstract- Methods’ that “A mixed-method open pilot trial design will be used” please note that any regression techniques are not basically/originally developed for any sort of [between or within group(s)] comparison(s). Moreover, a brief note on intervention designed may be expected/desirable/appreciated, by most of the readers, I guess. Except these few points, this manuscript is ‘excellent’ and therefore only ‘Minor revision’ is recommended.

Response: Thank you for your thorough review of the statistical aspects of this study. Thank you also for sharing the relevant extract from Medical Research Methodology.

We agree that it was overstated to assume we could get fair idea of attrition rates and yield/produce an effect size estimate to design a fully powered randomised controlled trial with the small sample size. 

In line with the ORBIT model (Czajkowski et al., 2015), we have decided to reframe this study as a proof-of-concept pilot study that seeks to determine whether a more rigorous pilot testing with a randomised design warranted. The model also states that sample sizes in proof-of-concept studies can be small because clinical, not statistical, benefit is sought (page 16, lines 354-356). Furthermore, Czajkowski and colleagues state that ‘sample size calculations are unnecessary’ for proof-of-concept studies, so we have decided not to pre-determine sample size for this study (page 18, lines 356-357).

Due to the lack of control group, expected small sample size and consequent insufficient power to test for significance, we have decided to perform only descriptive analyses. This is now stated on page 19 (lines 435-437) ‘Due to the lack of control group, expected small sample size and consequently insufficient power to test for significance, only descriptive analyses will be performed on the primary and secondary outcomes.’ And page 19 (lines 439-442): ‘Mean values and mean differences (MD) will be used to describe proof-of-concept for eating disorder psychopathology, %mBMI, depression, and anxiety from baseline and post-intervention. Data will be presented as either the percentage of change (for continuous variables) or absolute scores (for discrete variables).’

The limitations of the single-arm design, expected small sample size and use of only descriptive statistics only will be considered in future work. This is now stated on page 19 (lines 435-437) and page 21 (lines 503-508).

Thank you also for your suggestion to provide more detail on the intervention. As recommended by Reviewer 1, we have now described the intervention in accordance with the TIDieR checklist on pages 10-13, and have included the checklist as supplementary material.

---

## [Editor Report · Decision Letter 1]

19 Mar 2024

Guided self-help treatment for children and young people with threshold and subthreshold eating disorders: A pilot study protocol

PONE-D-23-42547R1

Dear Dr. Davey,

We’re pleased to inform you that your manuscript has been judged scientifically suitable for publication and will be formally accepted for publication once it meets all outstanding technical requirements.

Kind regards,

Simone Varrasi

Academic Editor

PLOS ONE
---

## [Editor Report · Acceptance letter]

1 Apr 2024

PONE-D-23-42547R1 

PLOS ONE

Dear Dr. Davey, 

I'm pleased to inform you that your manuscript has been deemed suitable for publication in PLOS ONE. Congratulations! Your manuscript is now being handed over to our production team.

Kind regards, 

on behalf of

Dr. Simone Varrasi 

Academic Editor

PLOS ONE